# Construction and Demonstration of the Evaluation System of Public Participation Level in Urban Planning Based on the Participatory Video of 'General Will—Particular Will'

**Zongxiang Wang** [1,2,*], **Tianhao Chen** [3], **Wei Li** [1,2], **Kai Zhang** [1,2] **and Jianwu Qi** [1,2]

1   College of Geography and Environmental Science, Northwest Normal University, Lanzhou 730070, China
2   Institute of Urban Planning and Tourism Landscape Design, Northwest Normal University, Lanzhou 730070, China
3   School of Architecture and Urban Planning, Suzhou University of Science and Technology, Suzhou 215000, China
*   Correspondence: 201775070130@nwnu.edu.cn

**Abstract:** Under the requirement of the modernization of the national governance system and governance capacity, it is an important measure for the government to respond to the demands of the public in the process of urban governance to explore more extensive and more universal means of public participation. With the advent of the Internet era, the communication method of using images as media has made public participation across time and space simple and convenient compared with the background, whereby the participation channels in past urban planning processes could not fully meet the public's demands. We Media, represented by participatory videos, has had a huge impact on public participation with the help of the widespread influence of the Internet. Using the political analysis framework of "general will—particular will", it is proposed that coordination between the cognitive level and the practical level is key to evaluate the level of public participation in participatory video intervention in urban planning. AHP and Delphi are used to build the index system. On the basis of adopting a comprehensive evaluation index, a coupled coordination model is introduced to build the public participation evaluation system of urban planning based on the participatory video of 'general will—particular will'. Through the evaluation of 4770 image samples and 200 survey materials from 11 communities in Xi'an, the index system is found to display good validity. Finally, from the perspective of different stakeholders, the implementation of participatory video intervention in public participation is summarized. This paper has important theoretical value and guiding significance in clarifying the impact of participatory video intervention on public participation in urban and rural planning and promoting the effective improvement of public participation in urban planning.

**Keywords:** coupling coordination degree; evaluation system; participatory video; public participation; urban governance

## 1. Introduction

In 2019, the Fourth Plenary Session of the 19th Central Committee of the Chinese Government incorporated the principle of co-governance and sharing into the category of 'social governance community', clearly put forward major propositions such as the 'modernization of the national governance system and governance capacity' and 'promotion of the downward shift of the focus of social governance and service to the grassroots', and they established a general plan of action for the construction of China's modern governance system in the new era [1]. Broadly speaking, public participation refers to 'all activities of citizens that influence public policies and public life' [2]. In terms of participation forms, they can be divided into online participation and offline participation, as well as

institutionalized participation and non-institutionalized participation. As a key link in urban planning, construction and management decision-making, public participation plays an unprecedented role in the process of urban development [3]. It is of great significance to explore people-centered urban construction and governance systems [4]. With the continuous development of China's modernized construction, the public is playing more and more attention to the individual's right to participate. However, the media and methods of public participation do not align with the awakening of public participation awareness and the rapid improvement in participation ability, resulting in a series of problems in urban governance, such as insufficient communication, poor communication, weak pertinence and planning of objectives, as well as realistic deviations. On the other hand, 'top-down' dominated public participation in urban planning lacks consideration of multi-subject dialogue, the media of resonance, and the means of sympathetic output.

The advent of globalization makes information dissemination and sharing the key to public consensus and conflict mitigation [5]. With the acceleration of digitalization and networking, video has gradually become the most important medium in building a common discourse. As the most intuitive and effective medium by which the public obtain information, video has great application and development potential in terms of public participation in urban planning due to their huge advantages in the rapid dissemination of information. The popularization of network media and intelligent equipment and the vigorous development of various video platforms provide a path for the construction of a public subject discourse. From the perspective of urban governance, emphasizing communication, interaction, and multi-subject participation appears to be key for future planning [6].

This paper argues that public participation has gradually developed from the traditional 'static information dissemination' to the 'dynamic mode' of extensive interaction between the government and the public, and participatory video has played an important role. Through empowerment, participatory video—as an effective platform for extensive public participation—represents a constructive communication medium with the government, thus promoting the effective solution of related problems. Whether there is a good channel for public participation is a basic index to evaluate the level of public participation in social governance at present [7]. With the increasing complexity of urban governance, participatory videos are regarded as a new method and means to promote public participation and improve urban governance, as indicated in a series of studies related to urban governance [8,9]. The issues of whether participatory videos can be used as an observation method to deconstruct space and interpret social problems; and as an effective means to promote multi-subject dialogue, public participation, and targeted planning and design; and whether there is a dilemma of operation or practice all require further exploration. Using the political theory of 'general will—particular will', this paper presents an empirical analysis on the practice of public participation in Xi'an Textile City and explores the unique value and practical significance of participatory video in current urban governance. At the same time, it also provides a new perspective and path for future public participation in urban planning. Using the analytic hierarchy process, Delphi method, expert judgment matrix, and so on, we construct a Chinese ecological resettlement area farmers evaluation index system of sustainable livelihoods, using the coupled coordination degree model (CCDM) built based on the 'general will—particular will' participatory video level of public participation in the urban planning evaluation system; and we focus on 11 communities in Xi'an as an example to illustrate empirical index system of inspection and correction. This paper has important theoretical value and guiding significance in clarifying the impact of participatory video on public participation in urban and rural planning and promoting the effective improvement of public participation in urban planning.

## 2. Connotations and Practice of Participatory Video

Participatory video involves the public taking control of the camera, using video as a communicative tool. It is used to record the subjective demands of individual members of the public and provide new channels for public participation. It is also known as commu-

nity video, village video, or grassroots video. It is a process of action or a working method derived from Western practices that promote public participation [7]. It has been widely used in the research of Western human geography and urban planning in recent years. The earliest video attempt began with Don Snowden in Fugu Island, Newfoundland, Canada, aiming to empower the public through video participation and change in government policies. The research mainly focuses on the concept, function, and operation methods in terms of practical effects, focusing on empowering residents to express themselves, improving internal communication in the community and promoting community problem solving. Chris Lunch emphasizes that the community exhibition of participatory video is the key to participatory video, and the process of projection and discussion can achieve positive results. Cahill and Bradley state that participatory video can effectively record social injustices, demonstrate the voice of justice in action, and help to open a new dialogue between stakeholders. The research of Manon and Sha Di in a Nicaraguan community proved that the participatory video method is an effective means to create two-way communication channels, which can directly promote gender empowerment, learning, and innovation, and achieve the goal of community production and an economic environment [10]. Shweta Kishore noted that participatory video can foster an active and dynamic public, particularly through media participation and dialogue, to form an evaluative, participatory public [11].

In the early 1990s, participatory videos were introduced into China as an effective working method by the World Bank, the United Nations, and other institutions, and they were gradually applied in environmental protection, poverty alleviation, education, community governance, and other fields. Participatory video focuses on various problems in urban and rural development, focuses on vulnerable and marginalized groups, and promotes the bottom-up expression of democratic rights. By empowering the people to examine their concerns and find solutions, it promotes public expression and participation, community action and communication with relevant government departments, and it ultimately promotes effective solutions to urban governance problems.

In the past, urban governance required participants to respond to relevant claims in strict accordance with the participatory process, but it was often unable to achieve the desired results for various reasons. With the advent of the Internet era, video—as a medium of communication—has made public participation easy and convenient, so that urban planning and other public affairs can occur with more extensive public participation [12]. At present, due to the rapid development of the Internet, all forms of We Media have begun to be widely used, in which participatory video has gradually been used in the field of urban governance [13,14]. The combination of participatory video and urban governance actually uses a more cooperative and participatory video practice to forge community self-organization, self-identity, self-management, and self-development. This is of original significance for urban governance in China [13].

## 3. Construction of Public Participation Model Based on Participatory Video of 'General Will—Particular Will'

J.J. Rousseau, the forerunner of French thought, deconstructed the theoretical connotation of public opinion in his "*du contrat social*". He divided public opinion into three categories: private will, general will, and particular will. Rousseau believed that private will can be subdivided into individual will and group will; the collection of private will constitutes public will, focusing on private interests; on the basis of the common interests of all people, public will expresses the 'public ego', 'public personality', or "*der gemeinsamer wille*" of the people, representing justice [15]. Based on the above ideas, German philosopher Hegel further proposed that general will should not be only the "*der gemeinsamer wille*" of most people or all people (because it is changeable at any time), but should reflect a higher level of "*der allgemeiner wille*" [16]. Durkheim, a French sociologist, believed that the improvement of the production level means that the division of labor in modern society tends to be refined. The highly heterogeneous group composition of the public makes it

difficult to form a 'universal will', and what must exist is a contractual relationship bound by interests namely, 'organic solidarity' [17].

The theoretical explorations of Rousseau, Hegel, and Durkheim provide a multidimensional analytical perspective in deconstructing the public participation path of participatory video intervention in urban planning. In the process of participatory video intervention in public participation, the formation of public will has undergone four key stages. The first is the private collection stage: public empowerment and expression. As an effective means of public empowerment, participatory videos transfer the control of the lens to the public individual, so as to reduce the discourse power of group expression to the public itself. Through the expression and communication of public individuals on related issues, the concentration of public discourse power and the integration and focus of public opinions are completed, and collective thinking, judgment, and requirements on urban governance and other issues are formed. The second is the stage of public intention deconstruction: distinguishing public opinion from private. Although public will is focused on private interests, its appeal objectively has the dual attributes of 'private' and 'public'. Public demands expressed by individuals and groups will affect the achievement of public wills. The third is the stage of public private transformation: whether public will, including publicity, can be transformed into public will. Based on the current development status focusing on the refinement and heterogeneity of the social division of labor, the traditional 'process' participation mode is obviously unable to meet the actual needs of social development. However, even groups with common interests and aspirations do not necessarily constitute the 'universal will' or agree on common interests as expected.

The reason is that public participation needs time and energy, but there is uncertainty about whether the ideal effect can be achieved. Therefore, there is inevitably a large amount of 'free riders' behavior, leading to the dilemma of collective action [18]. Participatory video is of special significance to the construction of an interactive mechanism of 'organic unity' that triggers collective behavior because of its wide audience and the high efficiency of media communication. Fourth is the stage of implementation of public will: as the 'universal will' of public will. The coupling coordination degree between government policy and the public's general will determine the level of urban governance. The government's ruling concept, ability, management level, authority, and mode have an important impact on the implementation of public will. As the medium of benign communication between the government and the public and the means of extensive public participation, the empowerment and expression of participatory videos are important suggestions and supplements in the process of public participation in urban planning, so as to make the ideal public participation in urban planning possible and promote the effective solution of urban governance problems (Figure 1).

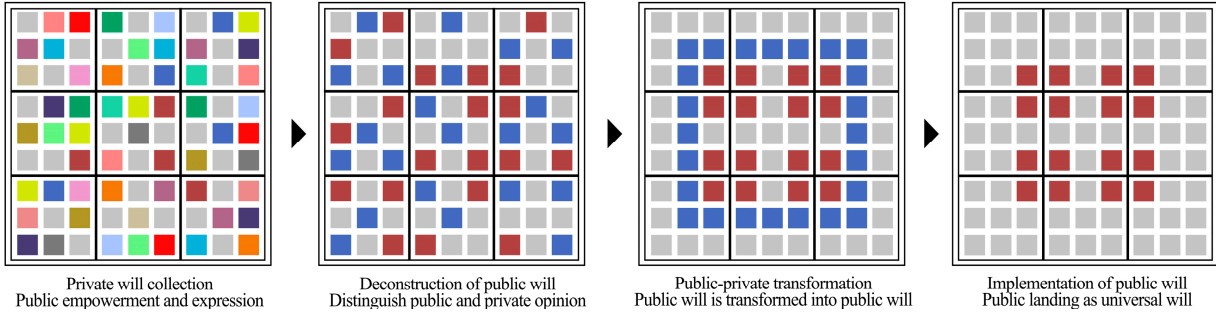

| Private will collection | Deconstruction of public will | Public-private transformation | Implementation of public will |
| Public empowerment and expression | Distinguish public and private opinion | Public will is transformed into public will | Public landing as universal will |

**Figure 1.** Practical pathway of participatory video intervention in terms of public participation in urban governance.

## 4. Construction of Public Participation Evaluation System in Urban Planning from the Perspective of Participatory Video

*4.1. Research Methods and Data Sources*

### 4.1.1. Research Methods

The analytic hierarchy process (AHP), also known as the multi-level weight analysis method, is a hierarchical weight decision analysis method proposed in the 1970s by applying network system theory and multi-objective comprehensive evaluation method. AHP mainly applies the system analysis method; the overall objective of the evaluation is to decompose the subject layer by layer to obtain each layer of the evaluation objective. The combination of qualitative and quantitative aspects results in a very effective system analysis method. The Delphi method is a type of consulting decision-making technology that can be widely used in various fields; it was proposed by the RAND Corporation in 1964. Through the objective aggregation of most expert opinions, probability estimation can be achieved for a large number of non-technical and non-quantitative factors. The core of this method is anonymous rounds of correspondence seeking expert advice. The organizer will summarize the opinions of each round and send them to each expert as reference materials for them to analyze and judge so as to put forward new argumentative opinions. This is repeated several times until the expert opinions converge and a more consistent and reliable conclusion or plan is obtained.

This paper uses AHP, the Delphi method, and an expert judgment matrix to construct the evaluation index system of the public participation level of urban planning from the perspective of participatory video. At the same time, as a multi-dimensional complex system, there are multiple internal coupling relationships between the practice and cognitive levels involved in public participation. Therefore, the study uses the coupling effect and CCDM to explain the relationship between practice and cognition. We take Xi'an Textile City as an example to carry out empirical testing and index system modification, providing a reference for the sustainable development of public participation in China.

### 4.1.2. Data Sources

In China, the unique national conditions of a large population and the humble and introverted national characteristics determine that most people have far fewer opportunities to express themselves than those who are represented. Regardless of differences in gender, ethnicity, and age, the public participates in social management and practice as 'representatives' most of the time. However, all individuals have a desire to express themselves and their opinions, and they have a legal right to do so. Participatory imaging permits extensive public participation and provides the possibility of expression and a voice for all groups in society. In order to deeply explore the current situation of public participation in urban governance in China, we select Xi'an Textile City as a case study. The evaluation of public participation in urban planning based on the 'public will—public will' participatory video mainly uses the participatory evaluation method to obtain case study and analysis data through a questionnaire survey, structural interviews, basic situation sampling, etc. From May 2022 to August 2022, the research team—together with the Textile City Neighborhood Office—carried out a study on participatory video intervention in urban planning and public participation, to guide the Textile City's public to focus on and feedback on regional urban governance issues. After three stages of shooting and sorting, the team processed 4770 effective image data of various platforms, including 1017 recorded image data and 3753 photos. Among them, the total video recording time was more than 35 h, 335 key shots were captured, and the photo data were classified into 15 categories of major urban governance issues, supplemented by 200 offline field interviews. In particular, 85.5% of the image data providers were residents of Textile City, and the proportions of people under 18 years old, 18–35 years old, 35–65 years old, and over 65 years old were 6%, 44%, 47%, and 3% respectively. In the participatory video proposal, prominent issues—such as the demolition of villages in cities, the reconstruction of old residential areas, the

construction of public green space, the renovation of public transport, and the governance of environmental sanitation—were presented (Figure 2).

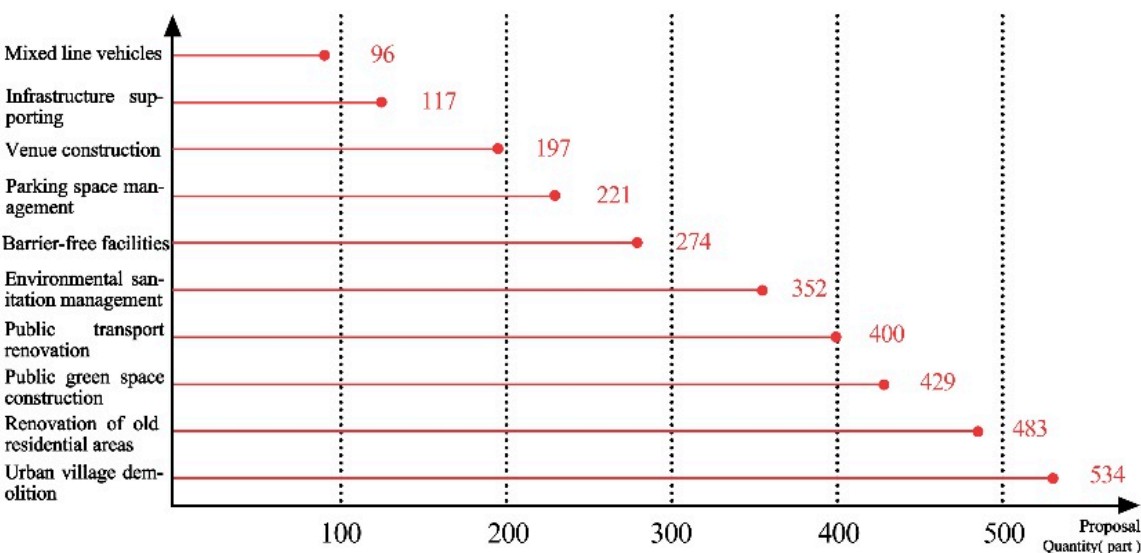

**Figure 2.** Types of participatory video proposals in Xi'an Textile City.

*4.2. Index Selection*

We used a variety of methods to select the evaluation index of the public participation level. Firstly, we studied and selected the indicators related to public participation in the United Nations Sustainable Development Goals (SDGs). The SDGs are a result-oriented framework for sustainable development—consisting of 17 goals, 169 targets, and 231 indicators—initially developed by the Inter-Agency Group of Experts on SDGs (IAEG-SDGs) [19]. The purpose is to encourage countries to use the framework to guide national planning, decision-making, and investment, and to regularly monitor and report on the progress of sustainable transformation from 2016 to 2030 [20]. It is the core concept and the axis principle that guides the economic and social development of all countries in the world [21]. Linking the practice of public participation in urban planning with the SDGs will help to improve the scientific accuracy and globality of the evaluation indicators, facilitate the realization of China's commitment to sustainable development, and promote the development process of China's urban planning and the dissemination of the practical experience of China's public participation. Secondly, the research group assessed the effectiveness, scientificity, homogeneity, identifiability, availability, and operability of each index through a field investigation of public participation in different areas under the participatory image intervention. Thirdly, the research group established a primary selection database of evaluation indicators by collecting and sorting academic research and government documents related to public participation at home and abroad. Finally, the research group invited different stakeholders—including scholars in the field of urban planning, representatives of government personnel, and residents participating in public discourse—to interviews in order to examine the development factors of public participation in urban planning through the involvement of different stakeholders. Finally, after the primary selection of the evaluation index system, the research group invited 10 scholars and government staff who had long studied public participation to discuss the scheme.

The research group used a five-point Likert scale and carried out an inquiry from 15 July to 25 July 2022 according to importance. Experts were invited to score the indicators according to their importance. Through the experts' index scores, the index was developed in consideration of the importance and coordination of the preliminary requirements. Furthermore, Kendall's Wa test was carried out on the expert scoring results, with $p = 0.000 < 0.05$I. It was proven that the expert scores had significant consistency and the

prediction results were reliable. Finally, an index system including 4 first-level indicators and 19 second-level indicators was formed.

### 4.3. Index Weight

Construction of Judgment Matrix and Consistency Check

In the evaluation system of the public participation level of urban planning based on the participatory video of 'general will—particular will', the effect and influence of each indicator are different, so it is necessary to assign weights to each indicator. Considering the actual situation of public participation and the complexity of the index system, this study adopted the analytic hierarchy process (AHP) in the subjective weighting method. After constructing the hierarchical structure model, it was divided into two stages to obtain the index weights of each layer. The first stage was to obtain the weight of the first-level indicator layer through the expert judgment matrix, and the acquisition process was divided into three steps. Firstly, in order to reduce the distortion of the evaluation results caused by human factors as much as possible, we used the Delphi method to solicit expert opinions and invited 10 experts to construct a pairwise comparison judgment matrix between the first-level indicators to determine the weight of each indicator. The weight was calculated using the following formula:

$$L_i = \sum x_{ij}/z \tag{1}$$

Here, $L_i \in (0,1)$, $x_{ij} \in (0,1)$, and $\sum L_i = 1$. $L_i$ is the weight of $i$ index, $z$ is the number of experts, and $x_{ij}$ is the score value of index $i$ by the $j$ TH expert. Since $L_i$ is an integer in the judgment matrix, the weight value will be expressed as an integer by rounding in the pairwise comparison between the judgment factors [22]. Referring to the weight calculation formula, we synthesized the scores of each expert and determined the judgment matrix for the first-level indicator based on this. Finally, Yaahp V12.7 software was used to calculate the weight of the first-level index layer by expert group decision-making, obtaining $C_i = 0.0454 < 0.1$. The consistency test of the judgment matrix was passed; that is, the first-level indicators of the urban planning public participation evaluation based on the participatory video of 'general will—particular will' were consistent, and the obtained weights were credible. The second stage aimed to further assign weights to the second-level indicators according to the average scores of experts (Table 1).

**Table 1.** Evaluation system of public participation level in urban planning based on participatory video of 'general will—particular will'.

| First-Level Indicator | Serial Number | Secondary Indicators | Weight |
|---|---|---|---|
| | | Cognitive Domain | |
| Consciousness of Participatory video (0.1953) | A1 | Participatory image identity | 0.1264 |
| | A2 | Acceptance of participatory video | 0.1561 |
| | A3 | Initiative of video participation | 0.3271 |
| | A4 | Video Participation Program Understanding | 0.3011 |
| | A5 | Completeness of video proposal | 0.089 |
| Video participation ability (0.1381) | B1 | Education level | 0.1977 |
| | B2 | Number of video platform participation | 0.2070 |
| | B3 | Video participation frequency | 0.2093 |
| | B4 | Tissue level of imaging intervention | 0.2000 |
| | | Practice Dimension | |
| Video participation Mode (0.3905) | C1 | Volunteering | 0.2319 |
| | C2 | Participatory video | 0.2667 |
| | C3 | Community participation | 0.2580 |
| | C4 | Network platform | 0.2435 |
| Video participation effect (0.2761) | D1 | Level of protection of public rights and interests | 0.1982 |
| | D2 | Community environment changes | 0.1843 |
| | D3 | Convenience of participatory video | 0.2120 |
| | D4 | Participatory video intervention effect | 0.2074 |
| | D5 | Public participation satisfaction | 0.1982 |

*4.4. Data Processing*

A Potential Problem

The evaluation of the level of public participation in urban planning based on the participatory video of 'general will—particular will' involves an index evaluation of public participation. The proposal of the evaluation index system must meet the management and evaluation requirements in terms of public participation by departments at all levels, so the practicability and operability of the index system must be given importance. In terms of data processing, there are a series of problems—such as the difference in magnitude and direction between different original data, and the difficulty of cross-year and cross-region comparison. Previous studies on index systems often used range standardization to process the original data, so as to make different indicators comparable across regions and years, and eliminate the differences in magnitude and direction among the original data. The formula for range normalization is as follows:

$$H_{ij} = \frac{N_{ij} - \min(N_{ij})}{\max(N_{ij}) - \min(N_{ij})} \tag{2}$$

Here, $H_{ij}$ is the standardized value of the $j$th index of system $i$; $N_{ij}$ is the original value of the $j$th index of system $i$; max $(N_{ij})$, min $(N_{ij})$ are the maximum and minimum values of the first index of the system, Min-max standardization was used to normalize the evaluation index data, and a zero value would appear after standardization. The zero-value index was processed by translation.

*4.5. Comprehensive Calculation*

4.5.1. Calculation of Comprehensive Evaluation Index

The multi-objective linear weighting function method was used to develop the evaluation system of the urban planning public participation level based on the participatory video of 'general will—particular will', and its function expression formula is

$$W = \sum_{i=1}^{m}(L_i \times E_i) \tag{3}$$

Here, $W$ is the system evaluation value; $L_i$ and $E_i$ are the standardized value and weight of the $i$th index, respectively.

The comprehensive evaluation index is $K$. In order to allow the comprehensive evaluation index to reflect the mutual relationship between public participation cognition and practice, and avoid reducing the range of $K$ values due to geometric weighting calculation, the comprehensive evaluation index $K$ in this study was calculated by arithmetic weighting, and its calculation formula is

$$K = \sum_{i=1}^{m}(Q_i \times W_i), \qquad \sum_{i=1}^{m} Q_i = 1 \tag{4}$$

Here, $W_i$ and $Q_i$ are the $i$th subsystem value and its weight respectively. In this work, $n = 2$, and the cognitive and practical aspects of the evaluation process are equally important, so Qcognitive = Qpractice and $Q_1 + Q_2 = 1$. Thus

$$K = Q_{cognitive} \times W_{cognitive} + Q_{practive} \times W_{practive} = \frac{1}{2}(W_{cognitive} + W_{practive}) \tag{5}$$

The comprehensive evaluation index $K$ is distributed within the range (0,1), which can be used to evaluate the comprehensive level of public participation in urban planning. The evaluation criteria are shown in Table 2.

**Table 2.** Classification criteria of comprehensive evaluation index, coordination level, and coordination development degree

| Section | W and K | R | J | Major Categories |
|---|---|---|---|---|
| [0, 0.1) | Very bad | Extreme maladjustment | Fading type | Type of maladjustment recession |
| [0.1, 0.2) | Bad | Severe maladjustment | Fading type | |
| [0.2, 0.30) | Very worse | Moderate maladjustment | Fading type | |
| [0.3, 0.4) | Worse | Mild maladjustment | Fading type | |
| [0.4, 0.5) | Very poor | On the verge of maladjustment | Fading type | Type of transitional development |
| [0.5, 0.6) | Poor | Grudging coordination | Type of development | |
| [0.6, 0.7) | Qualified | Primary coordination | Type of development | |
| [0.7, 0.8) | Good | Intermediate coordination | Type of development | Types of coordinated development |
| [0.8, 0.9) | Excellent | Good coordination | Type of development | |
| [0.9, 1] | Superexcellent | High quality coordination | Type of development | |

### 4.5.2. Coordinated Development Degree Calculation

The coordinated development degree is used to conduct comprehensive evaluation and research on the whole system. The standard formula of the coupling degree model commonly used at present is [23]

$$R = \left[ \frac{\prod_{i=1}^{n} W_i}{\left( \frac{1}{n} \sum_{i=1}^{n} W_i \right)^n} \right]^{\frac{1}{n}} \tag{6}$$

Here, $n$ is the number of subsystems; $W_i \in [0,1]$ is the value of each subsystem, with coupling degree $R \in [0,1]$. The larger the value of $R$, the smaller the degree of dispersion between practice and cognition, and the higher the degree of coupling; in the opposite case, the coupling degree is lower. The degree of coupling coordination is a comprehensive index of the degree of coupling and the level of development. It can reflect the coordination relationship between practice and cognition as well as the level of development. At present, the classification standard of coordination level defined by Liao Chongbin is used in most studies [24], and we deduced the division standard of coordinated development degree $J$ (Table 2). The CCDM constructed in this work is

$$J = \sqrt{R \times K} \tag{7}$$

## 5. An Empirical Study on Xi'an Textile City

### 5.1. Overview of Empirical Objects

Textile City is located in the east of Baqiao District, Xi'an City, Shaanxi Province. It is a typical industrial heritage residential area in China and also the political, economic, and cultural center of the Baqiao district. It covers an area of about 5.4 km$^2$ having a resident population of about 200,000 people. Xi'an Textile City is the largest light textile industrial production base in Northwest China, built during the "First Five-Year Plan". After the reform of state-owned enterprises in the 1990s, due to the adjustment of the state's industrial structure, the backwardness of its own production technology, market competition in coastal areas, and other factors, a large number of enterprises in the Textile City closed down, factories relocated, and employees laid off, with the area gradually becoming one of the poorest areas in the city. The textile city has 11 communities under its jurisdiction, namely Yiyin, Sanmian, Simian, Wuhuan, liumian, PowerChina, Fangyi, Fangxing, Xiangmin, Xiangyangfang, and Zaoyuanliu. A large number of industrial heritage sites, such as Soviet style factories and residential areas, are preserved in the region, including a certain scale of urban villages and old residential areas, and there are a large number of historical problems related to space property rights boundaries. The large population mobility, complex and diverse public demands, diverse community types and a complex personnel composition have brought great challenges to the local

urban governance. Based on this, the participatory video practice of Xi'an Textile City has explored the public participation path based on the 'general will—particular will' participatory video, with a view to widely soliciting public appeal, promoting an equal dialogue among multiple subject, and then promoting the implementation of the 'universal will' of the public, providing a new perspective for public participation.

*5.2. The Practice of Public Participation in Urban Planning in Xi'an Based on the Participatory Video of 'General Will—Particular Will'*

Participatory Video Intervention Design of Xi'an Textile City

Stage 1: Private collection stage. The public participation in participatory video intervention promotes the empowerment and expression of the public. In terms of participation methods, the team transfers the control of the lens to the public individual. Through media equipment such as mobile phones, the public marks the location of their concerns at any time and provides an image description, providing the public with the full space to express their views, realizing the reduction of the discourse power of group expression to the public itself and stimulate the initiative and enthusiasm of public participation. On the topic of participation, through the study and judgment of regional site problems, combined with the problems of public daily concern, mainly around the 'urban village demolition', 'public green space construction', 'environmental health governance', and other urban governance issues, we sought to carry out a public discussion, through the expression and communication of public individuals on related issues, to complete the concentration of public discourse power and the integration and focus of public opinions, forming collective thinking, judgment, and requirements with regard to urban governance and other issues.

Stage 2: The stage of public intention deconstruction. We perform a participatory video intervention, with the initial formation of public multi-dimensional governance of the site 'public opinion', through the initial classification of the public video, problem labeling, image narrative construction, and other measures to distinguish between the public's 'private' and 'public' demands.

Stage 3: Public–private transformation stage. The team organizes online meetings and forums on the online platform to promote communication and interaction between the public and the government and planners. At the same time, the independent public platform 'I Love Textile City', which is widely used by the public in the Textile City, is taken as the key delivery platform. The data—such as watching, reprinting, commenting, and commenting on public images—are taken as an important basis for the focus of public issues, and the stripping of public opinion is transformed into public opinion, which triggers collective behavior through an interactive mechanism.

Stage 4: Public implementation phase. Government departments assume core responsibilities in the implementation stage of public will, and the whole process of the project is jointly promoted by the team and the Textile City Street Office. By analyzing the feedback information of different platforms, the public's 'public will' is classified into 10 types of urban governance problems, and the public opinion is guided moderately to ensure the implementation of the public's 'general will' in Textile Cities.

According to the analysis of the image content (Figure 3), in terms of the demolition of villages in the city and the transformation of old residential areas, because the Textile City involves complex property rights issues, the residents of the Textile City hope that the relevant departments can send teams to settle in as soon as possible, clarify the ownership of property rights, and complete the signing of contracts as soon as possible. In terms of public green space construction, the residents of Textile City suggest that the regional space should be comprehensively renovated, more green plants should be added, landscape pieces should be added, and the living environment should be improved. As the aging phenomenon of the Textile City is prominent, residents also point out that barrier free facilities should be built, infrastructure supporting facilities should be improved, and the quality of life of the elderly should be improved. In terms of public space, the management of public parking spaces is the most prominent issue. In addition, some residents put

forward a number of suggestions, such as rectifying the problem of mixed traffic of people and vehicles in the three cotton and four cotton communities (Table 3).

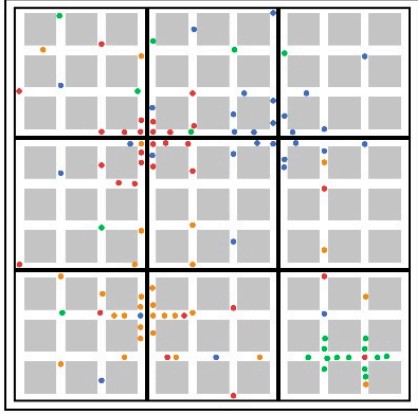

Image implementation unit with street as scale

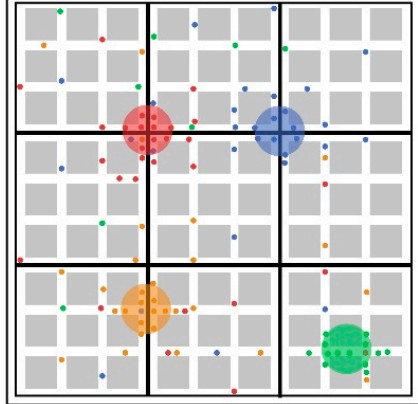

Public participation in Participatory video intervention

**Figure 3.** Indications of participatory video site intervention.

**Table 3.** Practical pathways of participatory video intervention in public participation.

| Stages | Steps | Specific Content |
|---|---|---|
| Private will collection | Personnel identification, site reconnaissance | Determine the image platform maintenance, local organization personnel, site reconnaissance, determine the place of image intervention. |
| | Platform determination and opinion collection | Select a wide spread of site platform, combined with public daily concerns, widely solicit opinions. |
| | Participatory image intervention | Identify participatory video intervention plans and identify modalities, platforms, and topics for participation. |
| | Problem focus | Integration and focus of public views through construction and analysis of participatory images. |
| Deconstruction of public will | Video classification, problem label | Organize participatory images of different platforms to distinguish public 'private' and 'public' demands. |
| Public–private transformation | Subject determination | Identify the theme after integrating and focusing public views. |
| | Community interaction | Detaching private will into public will, triggering collective behavior through interactive mechanisms. |
| Implementation of public will | Drawing up a plan | Organization planning units of government departments prepare planning schemes. |
| | Planning responses | Use participatory video analysis of urban governance issues and public demands for guidance and implementation planning. |
| | Offline negotiation | Invite stakeholders to explore landing priorities. |
| | Project implementation | The government departments cooperate with the planning units to jointly implement the planning scheme. |

### 5.3. Analysis of Empirical Results

Evaluation Result and Influencing Factor Analysis of Public Participation Level in Urban Planning of Xi'an City Based on Participatory Video of 'General Will—Particular Will'

In general, the public participation level of urban planning of Xi'an Textile City under the participation of participatory video is rated highly. The cognitive level index is 0.881–0.526, and the practical level index is 0.895–0.702. The overall level of practice and cognition is relatively high. The comprehensive evaluation index of public participation in urban planning in Xi'an based on the participatory video of 'general will—particular will' is 0.863–0.619, among which the comprehensive evaluation index of public participation in the Yiyin, Xiangyangfang, and Zaoyuanliu communities is excellent, and the comprehensive evaluation index of public participation in five communities San Mian, Si Mian, power

construction, textile medicine, and Xiangmin is good. The comprehensive evaluation index of the public participation level of three communities—including Fangxing—is qualified, and the comprehensive evaluation index of the public participation level of three communities—Yiyin, Xiangyangfang, and Zaoyuanliu—is excellent (Table 4).

**Table 4.** Comprehensive evaluation and coordinated development type of public participation in urban planning of Xi'an Textile City.

|  | Yiyin | San Mian | Si Mian | Wuhuan | Liumian | Dianjian |
|---|---|---|---|---|---|---|
| K | 0.847 | 0.790 | 0.762 | 0.688 | 0.688 | 0.752 |
| J | 0.846 | 0.790 | 0.762 | 0.685 | 0.687 | 0.749 |

|  | Fangyi | Fangxing | Xiangmin | Xiangyangfang | Zaoyuanliu |
|---|---|---|---|---|---|
| K | 0.702 | 0.619 | 0.734 | 0.821 | 0.863 |
| J | 0.701 | 0.612 | 0.733 | 0.821 | 0.863 |

Through the evaluation of the level of public participation in the urban planning of Xi'an Textile City under the intervention of participatory videos, the influencing factors are analyzed three levels: index weight (coefficient), actual development (independent variable), and coordination relationship (dependent variable).

(1) The index weight of each subsystem formed by the expert judgment matrix can be regarded as the stability coefficient of the public participation level of Xi'an urban planning based on the participatory video of 'general will—particular will', which is an objective judgment formed by experts in the field of urban planning based on their own long-term practice and research experience, and it will not change due to the evaluation object. In the first-level index layer, the three single index factors most affecting the level are the image participation mode (0.3905), image participation effect (0.2761), and image participation consciousness (0.1953). In the secondary indicator layer, the three major single indicator factors at the practical level are participatory imaging (0.2667), community participation (0.2580), and network platform (0.2435). At the cognitive level, the five major single index factors affecting the level are in as follows: the initiative of image participation (0.3271), the understanding of the image participation process (0.3011), and the frequency of image participation (0.2093).

(2) The actual development difference of each evaluation index of the urban planning public participation level based on the participatory video of 'general will—particular will' is the main factor affecting the evaluation of the urban planning public participation level, which can be understood as an independent variable of the development and evolution of urban planning public participation level. The evaluation of public participation in 11 communities of Xi'an Textile City shows that the internal strengths and weaknesses of the same evaluation object at the practical and cognitive levels are different. For example, the public education level and development level in the communities of Yiyin and Sanmian are the best, while the acceptance of participatory videos among public teams in the communities of Simian and Wuhuan is the best.

(3) The coordinated relationship between dependent variables at the reference practice and cognitive levels is an important factor affecting the coordinated development type of the public participation level in urban planning. The level of public participation in urban planning calculated by the comprehensive development index, which only considers the arithmetic mean of the practical and cognitive levels, cannot fully reflect the actual level of public participation in urban planning with participatory image intervention. Therefore, the coordinated relationship between dependent variables at the practical and cognitive levels significantly affects the evaluation of the final coordinated development type, which can effectively compensate for the inadequacy of only calculating the comprehensive evaluation index, and it can more effectively reflect the core idea of coordinated development at the practical and cognitive levels of public participation in urban planning.

## 6. Discussion

The practice and evaluation of Xi'an Textile City's participatory video intervention in urban planning public participation show that participatory videos based on 'general will—particular will' are effective in practical work as an effective means of public participation. This topic is analyzed from the perspective of different stakeholders, such as public participation subjects, planning preparation agencies, and planning management agencies.

### 6.1. Public Participation in Participatory Videos: Practical Significance in Urban Governance

In the process of urban planning compilation, the role of public participation has been paid ever more attention [25]. Whether there is a complete public participation system is an important basis for judging the legitimacy of planning decisions in the process of urban planning compilation. With the standardized development of urban planning, improving the level of public participation has become one of the important measures to promote the specialization, democratization, and scientificity of urban planning [26]. The introduction of participatory videos in the process of urban planning compilation is an innovative expansion and application of urban planning compilation and image technology. Participatory videos transfer the lens to community residents. Through the independent participation of community residents, they can independently control the lens so that residents have the opportunity to communicate and express their demands to the outside world, which is of great significance to further understanding the needs of residents and increasing bottom-up cognition [27].

The grassroots, immediacy, interactivity, and diversity of We Media greatly compensate for the disadvantages of the general public in information dissemination compared with high-level departments and elite groups—such as the government and experts—and it has gradually become an important tool for them to express their own interests. Participatory videos create a platform for public opinion expression, making it possible for more extensive public participation to come into reality. This type of community group participation and independent voice image participation in the preparation process allows community members to creatively express their subjective opinions, feelings, and ideas on different issues that are in line with their own interests through interviews and records of participatory video. This not only increases the participation and voices of different groups in society, but it also stimulates the enthusiasm of the public to participate in planning [28].

### 6.2. Public Empowerment: The Intrinsic Value of Participatory Video

Clay Sherky suggested that when people with different knowledge backgrounds and disposable time gather together for knowledge sharing, there will be considerable social effects [20]. In participatory video practice, the control of the lens is given to the public individual, and in fact the discourse power of group expression is given to the public itself. Compared with the traditional top-down urban governance process, participatory video practice has not only completed the optimization of urban governance procedures, but also allowed the concentration of public discourse power and the transfer of public opinions, which reflects a new creation and discussion of image communication mode. Participatory video seems to be a means of expression, but it is in fact a channel or measure, tool, and means for the public to obtain empowerment. Through self-participation and image expression, the public takes the initiative to present their own demands, and they engage in communication and exchange in the social domain so as to promote the relevant management departments to properly address the focus of public attention. The empowerment mentioned in participatory videos mainly focuses on the unique identity of public individuals. Empowerment does not entail manipulating the government through public discourse power, or coercing people with opinions, but should be defined as one of the processes of public participation. Through effective communication and expression, access to discourse and the action of the dominant groups and region-related independent individuals, it is possible to address relevant issues [29].

Participatory videos through public empowerment—through the individual expression and exchange of related issues, and through the integration of the individual point of view—lead to collective thinking and judgment of related issues. This empowerment is an important and valuable suggestion and effective supplement in the process of urban governance, so that the ideal urban planning public participation becomes possible. Social justice based on human equality, dignity, and rights is one of the core objectives of urban governance [30]. Participatory videos offer the public a greater opportunity to speak. In the practice of participatory videos, a video—as an effective means to empower the people and promote public participation—allows individuals to become involved on the basis of self-expression as a core identity, thinking about and discussing related issues from the first-person perspective. Through empowerment, participatory videos—as an effective platform for extensive public participation—form a medium for benign communication with the government, thus promoting the effective solution of related problems.

*6.3. Public Advocacy in Participatory Videos: The Action Orientation of Urban Governance*

In China, the unique national conditions of a large population and the national characteristics of humility and introversion determine that most people have far fewer opportunities to express themselves than represented. Regardless of their gender, ethnicity, and age differences, most of the time, the public are a 'represented' identity that participates in social management and practice. However, all individuals have the desire to express themselves, to communicate their views and exercise the power of speech given by law. Everyone should be an independent individual, a representative of themselves, and put forward their own views and express their own voice. Participatory videos bring a wide range of public participation, provides the possibility of expression and a voice for various groups of society—within a certain range—and alleviate the plight of some vulnerable groups marginalized by mainstream society [30]. As a form of relatively 'accessible' visual art, images can break the elite monopoly of art to some extent, help marginalized groups to accumulate cultural and social capital, and become a means for marginalized groups to build their own subjective spiritual space so as to realize the shaping and production of the social space [31].

Public advocacy under the guidance of participatory video allows the public individual to become the plenipotentiary of the self, through the lens of participatory presentation and the expression of individual or public demands [32]. The public can be better understood by society through expressing their demands and transmitting their voices to society. Therefore, participatory video is the action direction and working mode advocated for by the public. This participatory video supported by the public has strong influence and good social benefits [33].

Participatory videography is not only a powerful research tool, but it can also effectively enhance the group ability and internal and external communication. Under the lens of participatory video, the public can communicate with each other and discuss and think with each other by expressing their own views [34]. The group composition of the public represents different perspectives. Through the medium of participatory video, all groups express their demands equally, enhancing cognition and creating a unified consensus. This type of image-based communication and expression yields valuable opinions in the process of urban planning and even social development, which has an important guiding and enlightening effect on the direction of governance and control measures.

In recent years, with the deepening of the modernization of China's national governance system and governance capacity, there has been a new trend between the government's top-down social governance and the public's bottom-up social participation [35]. Planning is important among urban management institutions to accelerate part of their work with the Internet in order to build a more extensive channel for public dialogue—such as WeChat, microblogging, quivering, Facebook, and other media platforms—to achieve cross-temporal interaction. However, due to the constraints of various factors, there are still a series of problems in public participation in urban and rural governance in China—

such as insufficient public participation and insufficient communication between multiple subjects—which often leads to the 'bottom-up' expression of individual demands by the public through various channels, and this maintains the status quo, triggering group events related to network rights protection. This is also the opportunity for video-based media platforms to play an important role [36,37].

To promote the development and transformation of public participation requires the continuous summary of theory and practice. The application and development of participatory video is both an opportunity and a challenge [38]. If we cannot adapt to the rising demand of public participation and achieve the transformation of the participation mode in the Internet era—and promote the original, more symbolic public participation mode—we may intensify the misunderstanding and contradiction between the public and the government in the implementation of some measures [39]. In this study, with the use of the 'general will—particular will' political analysis framework, we select Xi'an Textile City to study the participatory video practice path of public participation. Finally, the implementation of a participatory video intervention in public participation is summarized from the perspective of different stakeholders—such as urban governance participants, planning institutions, and urban management institutions. Integrating participatory video into the public participation path of urban planning, guiding the public to express their own demands, and integrating and gathering public opinions will provide a new development path to address China's current lack of guidance and often extreme 'bottom-up' public participation. In terms of participants, they should conform to the trend of increasing public participation demands, seeking to empower the people and provide the public with the full space to express their opinions. In the process of participation, emphasis should be placed on building a long-term mechanism for equal dialogue between the public and planning and management institutions. Only when public participation in the new era complies with current trends of development and actively embraces the participation of new ideas and technologies can urban governance move toward the future of constructive cooperation and sustainable development [40,41].

Since the introduction of participatory video in China, with the continuous exploration of its theory and practice in the academic community, it has gradually matured in recent years [42]. It is expected that participatory video will be implemented more widely in future urban planning, so as to enrich the theoretical system of participatory planning in China and promote the development of urban governance.

## 7. Conclusions

This paper mainly constructs an evaluation system of the public participation level in Xi'an for urban planning based on participatory video accounting for the 'general will—particular will' dialectic. First of all, through community visits and surveys, it is proposed that the relationship between practice and cognition is the key to the evaluation of public participation level of image intervention in urban planning. We use AHP and Delphi to build an evaluation system including four first-class indicators—including image participation consciousness and image participation ability—and 19 s-class indicators. The expert judgment matrix and Delphi method are used to determine the weights of the primary and secondary indicators of each subsystem, and the data collection, scoring, and standardization methods of the secondary indicators are constructed. The comprehensive evaluation index $K$ is used to evaluate the level of public participation. At the same time, in order to better evaluate the public participation level of Xi'an Textile City under the intervention of images, the CCDM is further introduced to build an evaluation system of Xi'an's urban planning public participation level based on the 'public will—public will' participatory image. Finally, through the actual evaluation of 11 communities in Xi'an Textile City, the validity of the evaluation system is verified, and the index system is further revised.

**Author Contributions:** Conceptualization, Z.W.; Software, Z.W.; Validation, Z.W. and T.C.; Formal analysis, T.C.; Investigation, W.L.; Data curation, W.L.; Writing—original draft, K.Z.; Writing—review & editing, J.Q. All authors have read and agreed to the published version of the manuscript.

**Funding:** This work was supported by the National Social Science Foundation of China (approval no.:21bgl257) and the 2021 postgraduate research funding project of Northwest Normal University (approval no.:2021kyzz02142).

**Institutional Review Board Statement:** Not applicable.

**Informed Consent Statement:** Informed consent was obtained from all subjects involved in the study.

**Data Availability Statement:** Data will be made available on request.

**Conflicts of Interest:** The authors declare that they have no known competing financial interests or personal relationships that could have appeared to influence the work reported in this paper.

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
