# Peer review of "Construction and Demonstration of the Evaluation System of Public Participation Level in Urban Planning Based on the Participatory Video of ‘General Will—Particular Will’"

_sustainability, doi:10.3390/su15021687_

Round 1
Reviewer 1 Report (Previous Reviewer 1)
The article has been enhanced and the authors have adhered to the comments.
Author Response
Thank you very much for your careful and detailed review of the article. Your suggestions on the article have greatly improved the quality of our article.
On the other hand, we continue to optimize the references of the article and make some modifications to the language of the article.
Thanks again for your detailed review of the article! I wish you success in your work and good health!

Reviewer 2 Report (Previous Reviewer 2)
However the manuscript need some correction, but it could be published in present for
Author Response
Thank you very much for your careful and detailed review of the article. Your suggestions on the article have greatly improved the quality of our article.
On the other hand, we continue to optimize the references of the article and make some modifications to the language of the article.In the discussion section, we further cited the existing achievements, mainly focusing on articles 25, 26, 30, 33 and 41 of the references.
Thanks again for your detailed review of the article! I wish you success in your work and good health!

Reviewer 3 Report (Previous Reviewer 3)
Authors have addressed the comments satisfactory. No more comments from my side.
Author Response
Thank you very much for your careful and detailed review of the article. Your suggestions on the article have greatly improved the quality of our article.
On the other hand, we continue to optimize the references of the article and make some modifications to the language of the article.
Thanks again for your detailed review of the article! I wish you success in your work and good health!

Reviewer 4 Report (Previous Reviewer 4)
All comments are well addressed, this paper can be accepted for publication.
Author Response
Thank you very much for your careful and detailed review of the article. Your suggestions on the article have greatly improved the quality of our article.
On the other hand, we continue to optimize the references of the article and make some modifications to the language of the article.
Thanks again for your detailed review of the article! I wish you success in your work and good health!

This manuscript is a resubmission of an earlier submission. The following is a list of the peer review reports and author responses from that submission.
Round 1
Reviewer 1 Report
This is an interesting research area for public participation level in urban planning.
The abstract is thoughtfully produced and contains all required elements. However, the author could highlight the issue or provide a problem statement. The research problem statement has some relation to the problem, but is either incomplete, poorly expressed or tangential; the research problem should be clearly specified and should show significancy to the recent research on the area. There is lack of justification on the importance of the research issue.
The selection of research design and methods is appropriate however, it is not clearly presented; Research methods and their appropriateness for the research problem are reasonably explained, data production and analysis are explained but more justification is needed.
The discussion sensibly follows from the findings with some attempt to relate them to the broader context specified earlier in the text; The discussion uses the evidence/findings to address the research problem, question and objectives which are adequately addressed. Few sources were cited under discussion section, the author must include/addition of references to support the claims.
The conclusion explicitly relates key points in the evidence/findings to the broader context, with appropriate reference to the literature. However, some parts of the conclusion should be moved to the discussion section.
There are several typo errors in the manuscript. Proofreading is required.
Author Response
Response to Reviewer 1 Comments
The author would like to thank you for your detailed review of the article.At the same time, I strongly agree with the proposed amendments. Next I will complete the modification item by item.
Point 1: The abstract is thoughtfully produced and contains all required elements. However, the author could highlight the issue or provide a problem statement. The research problem statement has some relation to the problem, but is either incomplete, poorly expressed or tangential; the research problem should be clearly specified and should show significancy to the recent research on the area. There is lack of justification on the importance of the research issue.
Response 1: Thank you very much for your review of the manuscript. These opinions are very helpful for the revision and improvement of our paper, and also have important guiding significance for other studies. In the abstract part, the original text on the research problems and research significance of the elaboration is more vague. The author fully adopted the reviewer's comments and made supplementary revisions to the manuscript. The specific modifications are as follows:
The abstract complements the research questions and significance of the paper (lines 10-13 and 26-29 are shown in red).
Point 2: The selection of research design and methods is appropriate however, it is not clearly presented; Research methods and their appropriateness for the research problem are reasonably explained, data production and analysis are explained but more justification is needed.
Response 2: The original manuscript did not give a reasonable explanation for the selection of methods, and did not give sufficient reasons for the analysis of data. This is a major oversight of the paper. Thank you very much for raising this question, so that we can better improve the paper. The specific modifications are as follows:
In order to highlight this part more reasonably, we have made a major adjustment to the structure of the article. In the first section of the fourth part of the paper, 4.1 Research methods and data sources are clearly added.
In 4.1.1 Research Methods, the selected research methods and their applicability to research problems are introduced in detail. In 4.1.2 Research sources, the reasons for selecting cases for data analysis are analyzed. (Lines 212-264 are shown in red).
Point 3: Few sources were cited under discussion section, the author must include/addition of references to support the claims.
Response 3: The citation of references is a very important academic content. We fully agree with your point of view. we focus on adding references to the discussion part to support our point of view. We add new references to the corresponding positions of the article.
Specific modifications are indicated in the paper, mainly focusing on references 2, 10, 11, 28, 29, 30, 31, 35, 37etc.
Point 4: The conclusion explicitly relates key points in the evidence/findings to the broader context, with appropriate reference to the literature. However, some parts of the conclusion should be moved to the discussion section.
Response 4: Some parts of the conclusion were transferred to the discussion section, with the following modifications: (lines 636- 675 are shown in red).
Point 5: There are several typo errors in the manuscript. Proofreading is required.
Response 5: After proofreading of the manuscript, a large number of sentence errors were found, and detailed modifications were made for these contents. Because there are too many contents, the discussion will not be carried out here, and the detailed modifications will be marked in the manuscript.
The above is our reply and modification to your detailed comments. Thank you again for your careful review and detailed suggestions!

Reviewer 2 Report
In current study (sustainability-1976191-peer-review-v1), the authors empirically investigated and analyzed: “Construction and demonstration of the evaluation system of public participation level in urban planning based on the Participatory video of "General will——Particular will "
The paper topic is interesting, but it’s not new! Also, the way that present it, is not good.
It’s not structured as an academic paper. There is also English or structural mistakes in some sentences that made them not to be understand well. The paper doesn’t have enough nobility and quality to be published in this Journal.
Author Response
Response to Reviewer 2 Comments
The author would like to thank you for your detailed review of the article.At the same time, I strongly agree with the proposed amendments. Next I will complete the modification item by item.
Point 1: Articles should be revised according to the structure of academic papers.
Response 1: Thank you very much for reviewing the manuscript. These comments are very helpful for revising and improving our paper. In order to optimize the structure of the paper into an academic paper, the author has fully adopted the comments of the reviewers and made major modifications to the structure of the paper. The specific modifications are as follows:
- In the discussion of Section 1, an explanation of public participation and its meaning in this article has been added. (Lines 42-45 and 72-77 in red font).
- In the fourth part, the chapter of research methods and data sources is added, focusing on the applicability of research methods, and analyzing the reasons for selecting case data. (Lines 213 to 264, red font)
- Revise the discussion and conclusion of the article. Adjust the conclusion to the discussion section. (Lines 636-675, red font)
- The citation of references is a very important academic content. In order to make the article more academic, we added some new references to support our views. We have added new references in the corresponding positions of the article. This paper points out the specific modifications, mainly focusing on references 2, 10, 11, 28, 29, 30, 31, 35, 37, etc.
Point 2: There is also English or structural mistakes in some sentences that made them not to be understand well.
Response 2: There are a lot of problems in the sentence of the article, thank you very much for raising this question, so that we can better improve the paper. After proofreading of the manuscript, a large number of sentence errors were found, and detailed modifications were made for these contents. Because there are too many contents, the discussion will not be carried out here, and the detailed modifications will be marked in the manuscript.
The above is our reply and modification to your detailed comments. Thank you again for your careful review and detailed suggestions!

Reviewer 3 Report
(1) Good paper on the subject with scientific and empirical analysis.
(2) However, the followings points need to be improved.
(a) Public participation - I think that authors need to give a brief explanation on the meaning of public participation and what it means in this paper in the discussions at section 1
(b) Reference - I think that authors need to show all relevant reference name and year in the text, particularly section 2 and 3
(b) Typing mistakes - there are some typing mistakes here and there. Recommend reading the paper carefully and correct all typing mistakes.
Author Response
Response to Reviewer 3 Comments
The author would like to thank you for your detailed review of the article. At the same time, I strongly agree with the proposed amendments. Next I will complete the modification item by item.
Point 1: Public participation - I think that authors need to give a brief explanation on the meaning of public participation and what it means in this paper in the discussions at section 1.
Response 1: Thank you very much for your review of the manuscript. These opinions are very helpful to the revision and improvement of our paper. The explanation of the concept of public participation greatly improves the completeness and importance of the article, and we fully accept your suggestions. Specific modification measures are as follows:
In the first section of the article, the meaning of public participation is explained, and the concept of public participation is defined in this paper. ( Lines 42-45 and 72-77 in red font).
Point 2: Reference - I think that authors need to show all relevant reference name and year in the text, particularly section 2 and 3
Response 2: The year and name of references are important sources of academic citations. Thank you very much for raising this question.
The reason why the information is not shown in sections II and III is that the author uses the reference citation format specified by the journal. This reference format is marked with numbers in the text, and the information of the reference is mainly displayed in the end as a whole. Therefore, the corresponding literature information is not provided in this paper.
Point 3: Typing mistakes - there are some typing mistakes here and there. Recommend reading the paper carefully and correct all typing mistakes.
Response 3: Thank you very much for your review of the manuscript. The author fully adopts the opinions of the reviewers and has revised the manuscript. After proofreading of the manuscript, a large number of sentence errors were found, and detailed modifications were made for these contents. Because there are too many contents, the discussion will not be carried out here, and the detailed modifications will be marked in the manuscript.
The above is our reply and modification to your detailed comments. Thank you again for your careful review and detailed suggestions!

Reviewer 4 Report
This manuscript presents an interesting study of “Construction and demonstration of the evaluation system of public participation level in urban planning based on the Participatory video of "General will——Particular will”.
Overall, it combines a nice array of data sources to constructs the evaluation system of public participation level in Xi'an urban planning based on the Participatory video of " general will—particular will ". Initially through community visits and surveys, it is proposed that the relationship between practice and cognition is the key to the evaluation of public participation level of image intervention in urban planning. Then used AHP and Delphi to build an evaluation system including 4 first-class indicators such as image participation consciousness and image participation ability and 19 second-class indicators. The expert judgment matrix and Delphi method are used to determine the weights of the primary and secondary indicators of each subsystem, and the data collection, scoring and standardization methods of the secondary indicators are constructed
This paper is well organized, has clear objectives and the drawn conclusions are coherent with the obtained results. The results are well presented and the conclusions are relevant for urban planning/development community. However, there are some aspects of the manuscript that I believe should be addressed by the authors before publication. Firstly, the text needs extensive editing to correct typographical, grammatical and spelling mistakes. Examples are too many to be included in this report, but I could select a few just from the manuscript:
1. The key-words should be alphabetically arranged
2. Please re-write this sentence for better understanding “As one of the most intuitive and effective media for the public to obtain information, the transmission and interaction characteristics of images make them have great application and development potential in the planning participation mechanism”.
3. This sentence needs correction. “From the perspective of urban governance, urban governance emphasizing communication, interaction and multi-subject participation is the key element for planning the future”. It needs to be re-written as “From the perspective of urban governance, that emphasizing communication, interaction and multi-subject participation to be the key element for future planning”
4. The grammer and words in the manuscripts need to be corrected.
5. Please correct this sentence accordingly, “Public Participation is the Basic Requirement of Multi-subject Participation in Social Governance”.
6. Line # 264 to Line 266 are the same sentences and repeated. The duplication must be removed.
7. Some of the references are having different formats. The authors must use a uniform style for referencing. Some of the journal names are abbreviated, while in the other places full names of the journals are written. Please correct them accordingly. For example the reference # 17, the journal name is abbreviated.
8. English editing is required. Some of the sentences are confusing.
9. Please correct this sentence and remove the duplicated words, “This paper uses the political theory of “general will—particular will” political theory to empirically analyze the practice of participatory video intervention in public participation in urban planning in Xi 'an Textile City, and explores the unique value and practical significance of participatory video in current urban governance and public participation”.
10. The sentences used in various paragraphs were found to be too long. It will creates difficulty in understanding for the readers. Please re-phrase them for easy understanding.
11. Please check the line # 72, “using the coupled coordination degree model built based on the " general will—particular will ". It may be written as coupling coordination degree model (CCDM). Because it is used throughout the manuscript with this name. Also the abbreviation can be used afterwards for easy understanding of the readers.
12. Line # 74, Please correct the name of “Xian”.
13. Please check line # 83-87 and correct it accordingly, “derived from Western practice, mobilizing bottom voice and promoting public participation [6]. It has been widely used in the research of western human geography and urban planning in recent years”.
14. Please references these sentences, “. The research of Manon and Sha Di in Nicaragua community has proved that participatory video method is an effective way to create two-way communication channels” Also the sentence below, “Shweta Kishore noted that participatory video can foster an active and dynamic public, particularly through media participation and dialogue to form an evaluative, participatory public”.
15. Please revise these sentences, “With the advent of we - media era, the mobile Internet as a medium of communication makes cross - regional public mobilization become simple and convenient.so that urban planning and other public affairs Wider public participation. At present, with the rapid development of the Internet and the wide application of we media, participatory video is applied to the field of urban governance[10-11]”.
16. Please used the full name of the word and its abbreviation at first appearance, for example Sustainable Development Goals (SDGs), then used the abbreviation afterwards.
17. Line # 183-187, please replace the sign [。] with fullstop.
18. Please double check the spelling of Delphi in line # 230.
19. Please correct this sentence, “It covers an area of about 5.4 km2 and has a resident population of nearly 200000”. It may be corrected as “It covers an area of about 5.4 km2 having a resident population of about 2 lac people”.
20. Please reference this sentences, “Participatory video brings a wide range of public participation, provides the possibility of expression and voice for various groups of society, within a certain range, alleviates the plight of some vulnerable groups marginalized by main-stream society”(Sultan et al. 2022).
· Sultan H, Zhan J, Rashid W, Chu X, Bohnett E. 2022. Systematic Review of Multi-Dimensional Vulnerabilities in the Himalayas. International Journal of Environmental Research and Public Health 19:12177. https://doi.org/10.3390/ijerph191912177.
21. Last but not the least, I suggest the authors should add some updated references. As it is observed that, some sentences are not referenced in the manuscript.

Author Response
The author would like to thank you for your detailed review of the article.At the same time, I strongly agree with the proposed amendments. Next I will complete the modification item by item.
Point 1: The key-words should be alphabetically arranged
Response 1: Thank you very much for your review of the manuscript. The author fully adopts the opinions of the reviewers and has revised the manuscript. The specific modifications are as follows:
Keywords: coupling coordination degree; evaluation system; participatory video; public participation; urban governance(Lines 31 to 32 in red font).
Point 2: Please re-write this sentence for better understanding “As one of the most intuitive and effective media for the public to obtain information, the transmission and interaction characteristics of images make them have great application and development potential in the planning participation mechanism”.
Response 2: I deeply agree with your suggestion of modification, and change the original sentence to:As the most intuitive and effective medium for the public to obtain information, image has great application and development potential in terms of public participation in urban planning due to its huge advantages in the rapid dissemination of information.(Lines 62 to 65 in red font)
Point 3: This sentence needs correction. “From the perspective of urban governance, urban governance emphasizing communication, interaction and multi-subject participation is the key element for planning the future”. It needs to be re-written as “From the perspective of urban governance, that emphasizing communication, interaction and multi-subject participation to be the key element for future planning”
Response 3: We think your revision of this sentence is very appropriate and accurately expresses the meaning of the original. We agree unconditionally to the amendment of this sentence. (Lines 69 to 71 in red font)
Point 4: The grammer and words in the manuscripts need to be corrected.
Response 4: Thank you very much for your review of the manuscript. The author fully adopts the opinions of the reviewers and has revised the manuscript. After proofreading of the manuscript, a large number of sentence errors were found, and detailed modifications were made for these contents. Because there are too many contents, the discussion will not be carried out here, and the detailed modifications will be marked in the manuscript.
Point 5: Please correct this sentence accordingly, “Public Participation is the Basic Requirement of Multi-subject Participation in Social Governance”.
Response 5: We further summarize the content of the original text as follows: whether there is a good channel for public participation is a basic index to evaluate the level of public participation in social governance at present. (Lines 78 to 80 in red font)
Point 6: Line # 264 to Line 266 are the same sentences and repeated. The duplication must be removed.
Response 6: Thank you for your detailed review of the article. I will explain this part,Nij represents the standardized value of the jth index of system i, and the other represents the original value. The two symbols represent different meanings. For the sake of distinction, we make the following modifications: Hij. (Lines 343 to 344 in red font)
Point 7: Some of the references are having different formats. The authors must use a uniform style for referencing. Some of the journal names are abbreviated, while in the other places full names of the journals are written. Please correct them accordingly. For example the reference # 17, the journal name is abbreviated.
Response 7: References are an important part of the article. We fully agree with your allegations about the wrong content in this part, and have revised the article item by item. The specific modifications have been noted in the manuscript, and we have specially revised the 17th(It is now Article 20 after modification) reference you proposed. ( Lines 789-791 in red font)
Point 8: English editing is required. Some of the sentences are confusing.
Response 8: Thank you very much for your review of the manuscript. After proofreading of the manuscript, a large number of sentence errors were found, and detailed modifications were made for these contents. Because there are too many contents, the discussion will not be carried out here, and the detailed modifications will be marked in the manuscript.
Point 9: Please correct this sentence and remove the duplicated words, “This paper uses the political theory of “general will—particular will” political theory to empirically analyze the practice of participatory video intervention in public participation in urban planning in Xi 'an Textile City, and explores the unique value and practical significance of participatory video in current urban governance and public participation”.
Response 9: We have revised the problem you pointed out by changing the original text to: “Using the political theory of " general will—particular will ", this paper makes an empirical analysis on the practice of public participation in Xi 'an Textile City, and explores the unique value and practical significance of participatory images in current urban governance.” ( Lines 90-93 in red font)
Point 10: The sentences used in various paragraphs were found to be too long. It will creates difficulty in understanding for the readers. Please re-phrase them for easy understanding.
Response 10: We believe that the question you raised is completely accurate. We have revised the introduction, methods and other chapters of the article, and the revised content will be submitted in the revised draft.
Point 11: Please check the line # 72, “using the coupled coordination degree model built based on the " general will—particular will ". It may be written as coupling coordination degree model (CCDM). Because it is used throughout the manuscript with this name. Also the abbreviation can be used afterwards for easy understanding of the readers.
Response 11: We fully agree with your suggestions on this issue. We have revised the article and added (CCDM) in this part of the article. And elsewhere in the article, replace the original text with the CCDM. ( Lines 97、233、404、689 in red font)
Point 12: Line # 74, Please correct the name of “Xian”.
Response 12: Thank you for your proof of this error. We have corrected this error. ( Lines 99 in red font)
Point 13: Please check line # 83-87 and correct it accordingly, “derived from Western practice, mobilizing bottom voice and promoting public participation [6]. It has been widely used in the research of western human geography and urban planning in recent years”.
Response 13: Thank you very much for raising this question. The original text can't really express our meaning. We rewrote lines 81-87. ( Lines 110-114 in red font)
Point 14: Please references these sentences, “. The research of Manon and Sha Di in Nicaragua community has proved that participatory video method is an effective way to create two-way communication channels” Also the sentence below, “Shweta Kishore noted that participatory video can foster an active and dynamic public, particularly through media participation and dialogue to form an evaluative, participatory public”.
Response 14: We have cited references to your questions, and the references cited are Article 10 and Article 11. ( Lines 766-769 in red font)
Point 15: Please revise these sentences, “With the advent of we - media era, the mobile Internet as a medium of communication makes cross - regional public mobilization become simple and convenient.so that urban planning and other public affairs Wider public participation. At present, with the rapid development of the Internet and the wide application of we media, participatory video is applied to the field of urban governance [10-11]”.
Response 15: Thank you for your careful review of the article. Your suggestions have greatly improved the quality of the article. We have rewritten this sentence according to your suggestions. ( Lines 147-152 in red font)
Point 16: Please used the full name of the word and its abbreviation at first appearance, for example Sustainable Development Goals (SDGs), then used the abbreviation afterwards.
Response 16: We fully agree with the question raised. The specific modification method is: use the full name of the word and its abbreviation when it appears for the first time, and use the abbreviation for all subsequent content. ( Lines 268 in red font)
Point 17: Line # 183-187, please replace the sign [。] with fullstop.
Response 17: Thank you again for your detailed review of the article. We have made changes to the above problems and checked other parts with the problem. ( Lines 274-275、 299、316、397in red font)
Point 18: Please double check the spelling of Delphi in line # 230.
Response 18: Thank you for your detailed review. We have made changes to this issue.( Lines 311 in red font)
Point 19: Please correct this sentence, “It covers an area of about 5.4 km2 and has a resident population of nearly 200000”. It may be corrected as “It covers an area of about 5.4 km2 having a resident population of about 2 lac people”.
Response 19: Thank you for your detailed review. We fully agree with your careful revision of this sentence, and we have adopted it. ( Lines 412-413 in red font)
Point 20: Please reference this sentences, “Participatory video brings a wide range of public participation, provides the possibility of expression and voice for various groups of society, within a certain range, alleviates the plight of some vulnerable groups marginalized by main-stream society”(Sultan et al. 2022).
Response 20: We believe that your proposal on this issue is completely correct, and we have cited the literature. The reference cited is Article 28. ( Lines 616,811-812 in red font)
Point 21: Last but not the least, I suggest the authors should add some updated references. As it is observed that, some sentences are not referenced in the manuscript.
Response 21: The citation of references is a very important academic content. We fully agree with your point of view. At the same time, combining with the suggestions of other reviewers, we focus on adding references to the discussion part to support our point of view. We add new references to the corresponding positions of the article.
Specific modifications are indicated in the paper, mainly focusing on references 2, 10, 11, 28, 29, 30, 31, 35, 37etc.
The above is our reply and modification to your detailed comments. Thank you again for your careful review and detailed suggestions!

Round 2
Reviewer 2 Report
However some corrections are done, but still is not well structured.